# SHP2: A Redox-Sensitive Regulator Linking Immune Checkpoint Inhibitor Therapy to Cancer Treatment and Vascular Risk

**DOI:** 10.3390/antiox14121388

**Published:** 2025-11-21

**Authors:** Silvia Fernanda López Moreno, Stefania Assunto Lenz, Bernardo Casso-Chapa, Angelica Paniagua-Bojorges, Jung Hyun Kim, Nicolas L. Palaskas, Kevin T. Nead, Venkata S. K. Samanthapudi, Gilbert Mejia, Oanh Hoang, Jonghae Lee, Steven H. Lin, Joerg Herrmann, Guangyu Wang, Syed Wamique Yusuf, Cezar A. Iliescu, Noah I. Beinart, Charlotte Manisty, Masuko Ushio-Fukai, Tohru Fukai, Pietro Ameri, Roza I. Nurieva, Michelle A. T. Hildebrandt, Keri Schadler, Efstratios Koutroumpakis, Sivareddy Kotla, Nhat-Tu Le, Jun-ichi Abe

**Affiliations:** 1Department of Cardiology, The University of Texas MD Anderson Cancer Center, 1515 Holcombe Boulevard, Unit 1057, Houston, TX 77030, USA; a01284817@tec.mx (S.F.L.M.); a01198035@tec.mx (S.A.L.); a01721194@tec.mx (B.C.-C.); a01658857@tec.mx (A.P.-B.); jkim42@mdanderson.org (J.H.K.); nlpalaskas@mdanderson.org (N.L.P.); gfmejia@mdanderson.org (G.M.); onhoang@mdanderson.org (O.H.); jlee47@mdanderson.org (J.L.); syusuf@mdanderson.org (S.W.Y.); ciliescu@mdanderson.org (C.A.I.); nibeinart@mdanderson.org (N.I.B.); ekoutroumpakis@mdanderson.org (E.K.); skotla@mdanderson.org (S.K.); 2Escuela de Medicina y Ciencias de la Salud, Instituto Tecnológico y de Estudios Superiores de Monterrey, Monterrey 64710, Nuevo León, Mexico; 3Department of Epidemiology, The University of Texas MD Anderson Cancer Center, Houston, TX 77030, USA; ktnead@mdanderson.org; 4Department of Pediatric Research, The University of Texas MD Anderson Cancer Center, Houston, TX 77030, USA; klschadl@mdanderson.org; 5Department of Radiation Oncology, The University of Texas MD Anderson Cancer Center, Houston, TX 77030, USA; shlin@mdanderson.org; 6Cardio Oncology Clinic, Division of Preventive Cardiology, Department of Cardiovascular Medicine, Mayo Clinic, Rochester, MN 55905, USA; herrmann.joerg@mayo.edu; 7Department of Cardiovascular Sciences, Houston Methodist Research Institute, Houston, TX 77030, USA; gwang2@houstonmethodist.org (G.W.); nhle@houstonmethodist.org (N.-T.L.); 8Institute of Cardiovascular Science, University College London, London WC1E 6BT, UK; c.manisty@ucl.ac.uk; 9Cardio-Oncology Department, Barts Heart Centre, Barts Health NHS Trust, London EC1A 7BE, UK; 10Department of Medicine, Cardiology, Vascular Biology Center, Medical College of Georgia, Augusta University, Augusta, GA 30912, USA; mfukai@augusta.edu (M.U.-F.); tfukai@augusta.edu (T.F.); 11Cardiovascular Disease Unit, IRCCS Ospedale Policlinico San Martino, 16132 Genova, Italy; pietroameri@unige.it; 12Department of Immunology, Division of Discovery Science, The University of Texas MD Anderson Cancer Center, Houston, TX 77030, USA; nurieva@mdanderson.org; 13Department of Lymphoma/Myeloma, Division of Cancer Medicine, The University of Texas MD Anderson Cancer Center, Houston, TX 77030, USA; mhildebr@mdanderson.org

**Keywords:** ICI, PD-1, PD-L1, SHP2 tyrosine phosphatase, scaffold protein, oxidative stress, cancer, and cardiovascular disease

## Abstract

Src homology 2-domain containing protein tyrosine phosphatase 2 (SHP2), encoded by the *Ptpn11* gene (Tyrosine-protein phosphatase non-receptor type 11), is a key downstream effector of PD-1/PD-L1 signaling and is likely important, in addition to immune modulation, in tumor development and vascular homeostasis. SHP2 conveys PD-1 mediated inhibitory signaling in T cells, and is emerging as a therapeutic target. Importantly, there is an association between immune checkpoint inhibitors (ICIs), immune-related adverse events (irAEs), and cardiovascular complications, underscoring the need to understand SHP2’s role in these processes. This review aims to summarize current knowledge on SHP2/*PTPN11* biology, its role in immune regulation, cancer progression, and vascular homeostasis, and to discuss emerging therapeutic strategies targeting this pathway. The concept of using SHP2 inhibitors with immune checkpoint inhibitors (ICIs) is being investigated to address ICI resistance and to improve anti-tumor efficacy substantially. SHP2 is also being studied in non-cancer cell contexts, and signaling responses can differ by large magnitudes depending on the biological context and stimuli. Under normal circumstances, SHP2 promotes vascular homeostasis in endothelial cells (ECs) and myeloid cells and inhibits inflammation, and the reduction in SHP2 activity by oxidative stress, such as in atherosclerosis or diabetes, upregulates inflammation. In contrast, in response to radiation, the fibrotic response and subsequent lung injury were increased by endothelial SHP2 induction via Notch-Jag1 signaling. Vascular smooth muscle cells SHP2 act as a pro-atherogenic effector by enhancing ERK/MAPK signaling, and the upregulation of mitochondria localized SHP2 can also induce cellular senescence-associated inflammation by upregulating mitochondrial reactive oxygen species. Taken together, the two opposite signaling effects of SHP2 suggest that both the immune and vascular system responses appear to be more modulated by the redox, cell, and compartment-specific signaling of SHP2. More studies are needed for mitigating cardiovascular toxicity to patients, particularly with ICI-based treatment regimens.

## 1. Introduction

Immune checkpoint inhibitors (ICIs) have fundamentally changed the way we think about cancer treatment, representing a paradigm shift in innovations that are still affecting the outcomes of cancer care today [1]. ICIs were initially developed to treat metastatic melanoma, non-small cell lung cancer, renal cell carcinoma, and head and neck cancers, and are still in use as all four of these tumor types comprise the majority of ICI use clinically. There are a number of key ICIs including PD-1/PD-L1 inhibitors, LAG-3 (Lymphocyte Activation Gene-3) inhibitors and TIM-3 (T cell Immunoglobulin, Mucin-domain containing-3) inhibitors, and CTLA-4 inhibitors. As co-inhibitory receptors, PD-1 and CTLA-4 are critical for regulating T cell-mediated immune responses for immunological tolerance but can also allow the immune system to attack healthy cells by regulating the immune response. The immune system’s regulatory capacity also contributes to tumor suppression. Immune checkpoint molecules such as PD-1 and CTLA-4, along with their respective inhibitors (anti-PD-1/anti-PD-L1 and anti-CTLA-4), enhance anti-tumor immunity by attenuating negative feedback mechanisms that normally restrain immune activation at checkpoint interfaces. Having shown promising efficacy for several cancers, the ICIs represent a relatively novel approach, offering a new line of defense for metastatic disease overall [2]. This promising therapy has its drawbacks as well; ICIs are associated with immune-related adverse events (irAEs) that affect a significant portion of patients that receive them. Studies have shown that 60–80% of people treated successfully with ICI treatments have experienced at least one irAE [3]. Cardiovascular complications have been listed as one of the most concerning of the irAEs. Based on a study involving Florida’s Integrated Data Repository with over 100,000 cancer patients, 14.6% of patients who had at least one ICI were diagnosed with a new cardiovascular disease after treatment, and the association of cardiovascular toxicity in regard to ICIs continues to be a growing concern [4].

This review will concentrate on the immunological and vascular effects of PD-1/PD-L1 inhibitors, with special focus on the downstream signaling mediator SHP2 (Src homology 2-containing protein tyrosine phosphatase 2) and its broad implications for both processes. SHP2 is a multifunctional phosphatase that accepts signals from disparate receptor systems and oxidative stress, resulting in immune regulation and vascular homeostasis. Although previously viewed as a functional positive regulator of receptor tyrosine kinase (RTK) signaling, SHP2 implements inhibitor and phantom relationships depending on the cellular context and receptor inputs. In the immune context, SHP2 is an effector molecule for PD-L1/PD-1 signaling involving T cell inhibitory elements to facilitate peripheral tolerance and to dampen T overactivation. More importantly, new mouse evidence has pointed to a role for selective SHP2 inhibition in myeloid cells that may inhibit tumorogenesis independently of the PD-1 inhibition pathway, thereby implicating SHP2’s immunoregulatory functions extend beyond classical checkpoint pathways, and reaffirms the rationale for further pursue pharmacological inhibition both in the context of cancer and inflammatory/oxidative stress conditions [5]. This article will examine the nascent understandings of SHP2 acting in immune regulatory functions and vascular homeostasis, and the potential repercussions in the decisions made for patient with ICI-induced cardiovascular disease.

## 2. Structural and Functional Characteristics of SHP2: Mechanisms of Activation and Regulation

Phosphorylation is an essential post-translational modification which controls intracellular signaling pathways, and the processes through which kinases and phosphatases regulate phosphorylation are important for cellular homeostasis [6]. SHP2, which is encoded by the *PTPN11* gene, is a non-receptor protein tyrosine phosphatase that is widely expressed and involved in the organization of intracellular signaling networks. SHP2 is part of the SHP family of phosphatases, which includes SHP1 (*PTPN6*). While these enzymes share 61% identity in sequence, their expression profiles and functions are very different [7]. SHP1 is confined to hematopoietic and select epithelial cells where it is typically a negative regulator of receptor-mediated signaling [8]. On the other hand, SHP2 is expressed in nearly all tissues, and exerts both positive and negative regulatory effects, highlighting the pleiotropic functions of SHP2 in varying physiological and pathophysiological contexts [9,10].

SHP2 regulates cellular functions based on its enzymatic activity and functional interactions. As a multidomain protein, SHP2 consists of two tandem Src Homology 2 (SH2) domains (N-SH2 and C-SH2), a catalytic protein tyrosine phosphatase (PTP) domain, a C-terminal tail with two regulatory tyrosine phosphorylation sites (Y542 and Y580), and a proline-rich motif [11] (Figure 1A). SHP2 has a significant role in generating cellular communications and is a critical regulator of intracellular signaling. SHP2’s unique configuration allows for both catalytic phosphatase and scaffold protein to occupy a central position between numerous signaling cascades. By selectively dephosphorylating phosphotyrosine residues, SHP2 modulates the intensity and duration of downstream signaling events, allowing cells to appropriately respond to external signals [9]. The dual functionality of SHP2 is especially critical during receptor tyrosine kinase (RTK)-mediated pathways, where SHP2 helps transmit and propagate the signals of RTK such as epidermal growth factor receptor (EGFR), fibroblast growth factor receptor (FGFR), and platelet-derived growth factor receptor (PDGFR), which contributes to biological processes such as proliferation, differentiation, and survival [10]. SHP2 exhibits dual functionality: it not only binds to tyrosine-phosphorylated proteins but also possesses intrinsic tyrosine phosphatase activity. This two-step mechanism—initial binding followed by dephosphorylation—enables SHP2 to act as a dynamic regulator of signal transduction (Figure 1B). As we will elaborate later, the catalytically inactive (dominant-negative) form of SHP2 not only lacks phosphatase activity but also retains its ability to bind tyrosine-phosphorylated targets. This results in a “substrate trapping effect” that sequesters signaling intermediates and thereby inhibits downstream signaling (Figure 2A). Understanding the interplay between SHP2’s phosphatase activity and its binding to phosphotyrosine-containing proteins is essential for deciphering its complex regulatory roles in cellular signaling.

### 2.1. SHP2 as a Phosphatase and Scaffold Protein: Context-Dependent Regulatory Effects

Phosphatase enzyme activation depends on SHP2, which exists in a fragile balance between an open active conformation and an autoinhibited basal state [12]. The N-SH2 domain in the basal state limits access to the catalytic site contained in the PTP domain, therefore preventing substrate from engaging the PTP catalytic site. When SHP2 is stimulated with growth factors (e.g., EGF) or other cytokines, the phosphotyrosine residues of the bound molecules will bind to the N-SH2 domain, and surface interactions and conformations will induce release of this autoinhibition to expose the catalytic site and activate the SHP2 phosphatase activity [8,13] (Figure 1B). Also, the unique C-terminal tail of SHP2 is a significant regulatory region for SHP2 activity. As an example, the phosphorylation of Y542 would disrupt the N-SH2 domain’s interaction with the PTP domain, allowing an active conformation of SHP2 to occur. Interestingly, Y580 phosphorylation improved SHP2 catalytic efficiency by enhancing the engagement of the C-SH2 domain of SHP2 [9,13,14] (Figure 1B).

As a phosphatase, SHP2 is a fundamental component for intracellular signaling as it negatively directs the phosphorylation state of important adaptor proteins that contribute to signaling. Among the way phosphatases are classically defined as negative regulators of signaling, SHP2 is unique; SHP2 can activate signaling pathways with dephosphorylation by promoting the complex with proteins necessary to signal. The functional role of SHP2 in this regard is particularly important as it is a positive regulator of receptor tyrosine kinase (RTK) signaling, as well as the Ras/mitogen-activated protein kinase (MAPK) pathway [13]. In this role, SHP2 actively works on specific phosphotyrosine residues of the various adaptor proteins, including the single insulin receptor substrate protein, IRS1, and Grb2 associated binder 1 (GAB1) (Figure 1B and Figure 2). Both IRS1 and GAB1 have over-phosphorylated states that can sequester or recruit inhibitory proteins. By either dephosphorylating inhibitory phosphotyrosines or dephosphorylating non-productive phosphotyrosines, SHP2 promotes conformational changes or exposure of new docking sites for recruiting the GRB2/SOS protein complex to the plasma membrane [8]. GRB2/SOS is crucial for Ras proteins that are inactivated because SOS catalyzes the exchange of GDP for GTP on Ras, converting Ras to the active GTP-bound Ras. Activated Ras initiates a downstream kinase cascade (RAF → MEK → ERK), ultimately activating the transcription of the target genes responsible for proliferation, survival, and motility [8] (Figure 2A).

It has been reported that tyrosine-phosphorylated substrates can engage not only the SH2 domains of SHP2 but also its catalytic PTP domain (Figure 2B). Structural studies have demonstrated that the SHP2 PTP domain can directly bind phosphopeptides derived from IRS1 and CD28, extending beyond the classical SH2–phosphotyrosine docking mechanism [15]. Recent crystal structures further revealed that the PTP active site accommodates the phosphorylated tyrosine while simultaneously sensing adjacent Ser/Thr phosphorylation, which stabilizes substrate binding and modulates catalytic efficiency. This represents a form of substrate-assisted regulation, emphasizing that SHP2 substrate specificity and activity are shaped by the broader sequence context of its binding partners, rather than solely by SH2 domain recruitment [15]. Functionally, this dual mode of substrate recognition allows SHP2 to fine-tune signaling in diverse biological contexts. In insulin signaling, SHP2 binding and dephosphorylation of IRS1 help regulate the PI3K–AKT pathway, preventing excessive activation and maintaining balanced metabolic responses [16] (Figure 2B). In T cell activation, SHP2 interaction with phosphorylated CD28 modulates co-stimulatory signaling, acting as a checkpoint that defines activation thresholds as also described later [17,18] (Figure 2B). Collectively, these findings establish SHP2 as a critical signal integrator that is recruited through SH2 docking but exerts additional regulatory control through direct PTP–substrate interactions, and thus, shapes pathways essential to metabolism, immunity, and disease.

SHP2 is a further example of a positive modulator of AMP-activated protein kinase (AMPK), particularly in relationship to metabolic challenge (Figure 3). While SHP2 is typically discussed in the chemotherapy context of mitogenic signaling, it can also help with energy homeostasis via facilitating AMPK activation. The phosphatase activity of SHP2 decreases the inhibitory phosphorylation of AMPK-mTOR axis components, which initiates AMPK activation [19]. The overall outcome of AMPK activation is the inhibition of mTORC1 activity and stimulation of autophagy (particularly through ULK1 phosphorylation) [20]. For example, in pulmonary fibroblasts, SHP2 has been shown to mediate AMPK activation as a means to promote mitophagy and metabolic reprogramming during mitochondrial stress [19]. In a neural context, SHP2 acts as a positive upstream modulator of AMPK, a master regulator of cellular energy homeostasis. Functionally, SHP2 promotes AMPK activation under energy-deprived conditions, allowing AMPK to inhibit anabolic processes and maintain energy balance. AMPK, in turn, suppresses the mTOR/S6K1 (S6 kinase 1) pathway—a key driver of protein synthesis and cell growth—by phosphorylating upstream regulators like TSC2 and Raptor. Loss of SHP2 results in impaired AMPK signaling, leading to unchecked S6K1 activity and dysregulation of cellular growth responses [21,22]. While SHP2 is best known for its canonical pathways of signaling, the information presented above demonstrates that SHP2 functions as a metabolic integrator in animal cells to link energy sensing to adaptive cellular responses.

SHP2 is a positive modulator of AMP-activated protein kinase (AMPK), particularly under metabolic stress (Figure 3). Although commonly associated with mitogenic signaling in chemotherapy contexts, SHP2 also supports energy homeostasis by facilitating AMPK activation. Its phosphatase activity reduces inhibitory phosphorylation within the AMPK–mTOR axis, triggering AMPK activation [19]. Activated AMPK suppresses mTORC1 and promotes autophagy, notably through ULK1 phosphorylation [20]. In pulmonary fibroblasts, SHP2-mediated AMPK activation enhances mitophagy and metabolic reprogramming during mitochondrial stress [19]. Similarly, in neurons, SHP2 acts upstream of AMPK to maintain energy balance under nutrient deprivation by inhibiting anabolic pathways. AMPK then downregulates the mTOR/S6K1 pathway via phosphorylation of TSC2 and Raptor. Loss of SHP2 impairs AMPK signaling, leading to unchecked S6K1 activity and dysregulated growth responses [21,22]. These findings highlight SHP2 as a metabolic integrator linking energy sensing to adaptive cellular responses.

While SHP2 activates mitogenic pathways such as Ras/MAPK, it can negatively regulate JAK/STAT signaling (Figure 4). This occurs mainly through its phosphatase activity, which dephosphorylates STAT1, STAT3, and STAT5 after cytokine stimulation, preventing dimerization, nuclear translocation, and gene activation. In immune cells, this mechanism helps maintain homeostasis and limit inflammation. SHP2 deficiency prolongs STAT1 phosphorylation, increasing IFN-γ sensitivity and immune activation [23]. Conversely, SHP2 overexpression inhibits IL-3-induced STAT5 activation, reducing hematopoietic cell proliferation [24]. In prostate cancer, SHP2 downregulates STAT1, as its loss enhances STAT1 phosphorylation and immune marker expression (HLA-ABC, PD-L1) [25]. SHP2 also modulates IL-6 signaling by restraining excessive STAT3 activation while preserving pathway integrity [26]. Overall, SHP2 acts as a phosphatase that constrains JAK/STAT signaling, balancing immune responses and maintaining signaling fidelity across contexts.

Thus, SHP2 enhances signaling by changing the phosphorylation landscape to allow productive protein–protein interactions and downstream activation as opposed to inhibiting signaling. SHP2 can use dephosphorylation in a context-dependent manner, suggesting that SHP2 acts as a signal amplifier and not a signal terminator [13]. The precise and regulated activation of SHP2 is more pertinent in the context of the associated mutations in the *PTPN11* gene of SHP2. *PTPN11* gain-of-function mutations (e.g., Noonan syndrome) ablate the autoinhibited binding of the N-SH2 (N-terminal SH2) domain on the PTP, actively creating a situation where SHP2 is constitutively activated with complete freedom from having to rely on upstream signals. This aberrant activation is likely to “amplify Ras/MAPK signaling” and promote the various developmental features observed in patients with Noonan syndrome [27] (Figure 3). Mutations causing LEOPARD Syndrome work based on variations on the mutation and may often be catalytically impaired in the normal process but still allow for pathological SHP2 activation by destabilizing the inactivity conformation, alter binding specificity on the two possible SH2 domains, and enable greater opportunities to engage with signaling partners [28,29]. These studies are not only enlightening for determining SHP2 activity but also raise important points about the conformation and mechanisms of protein–protein interactions of SHP2 and regulation of intracellular signaling. These contexts also provide an important distinction of context-dependent effects; in some instances, these mutations obstruct phosphatase activity, while allowing for signaling through the mutation, causing a prominence of conformation or adapter roles [29].

SHP2 regulates PI3K/AKT signaling in a context-dependent manner through interacting with p85, the PI3K regulatory subunit (Figure 3). In growth factor signaling (e.g., EGF, IGF) and pharmacologic models, SHP2 acts as a positive regulator by promoting PI3K activation and subsequent AKT phosphorylation. This facilitation requires the N-terminal SH2 domain to bind p85 [30] (Figure 3). In oncogenic v-Src signaling, SHP2 is essential for AKT activation and survival signaling [31] (Figure 3). Conversely, in ovarian granulosa cells under oxidative stress (e.g., H_2_O_2_), SHP2 binds p85 but suppresses AKT phosphorylation, acting as a negative regulator [32] (Figure 4). These findings indicate that SHP2’s role in PI3K/AKT signaling varies by stimulus, cell type, and disease context, reflecting its dynamic function in cellular networks.

### 2.2. SHP2 in Mitochondria: Balancing ROS, Inflammation, and Senescence

The recent discovery of SHP2 in mitochondria has enhanced our understanding of SHP2 cellular function, specifically with respect to mitochondrial function and energy metabolism. SHP2 was first identified as a tyrosine phosphatase in the mitochondria and has contributed greatly to mitochondrial bioenergetics [33,34]. For instance, gain-of-function mutations in the *PTPN11* gene, E76K and D61G, increased the activity of the mitochondria complexes I and III electron transport chain, increased mitochondrial membrane potential, and increased reactive oxygen species (mtROS) production; all findings are indicative of high oxidative stress and metabolic activity. These examples, along with many others, highlight the important contribution of SHP2 to the delicate balance of energy production and redox homeostasis of the mitochondria [33] (Figure 5).

SHP2 has also been implicated in the regulation of innate immunity in regard to mitochondrion-derived immune responses. Importantly, SHP2 does not solely promote mtROS production; it can also act to suppress excessive ROS generation under certain conditions. One such mechanism involves the regulation of adenine nucleotide translocase 1 (ANT1), where SHP2-mediated dephosphorylation helps maintain mitochondrial membrane potential and prevents the release of mitoROS and mitochondrial DNA. Upon the activation of the NLRP3 inflammasome, it has been shown that SHP2 schedules translocation to the mitochondria to dephosphorylate ANT1, a protein that is part of the mitochondrial permeability transition pore (mPTP), which prevents loss of mitochondrial membrane potential, and thus prevents loss of mitochondrial DNA and ROS, which are both critical for the activation of the NLRP3 pathway. Therefore, dephosphorylation of ANT1 represents a key strategy for managing NLRP3 hyper-activation to limit overactive mtROS and inflammatory responses. These findings suggest that there is a potential for SHP2 to modulate mitochondria functions and thereby mitochondrion-derived immune signaling to regulate specific cellular stress responses to preserve mitochondrial integrity and limit inflammation damage [35] (Figure 5).

SHP2 is a central player in mitochondrial function, cellular metabolism, and inflammation—three interrelated pathways that directly influence cellular senescence. However, the role of mitochondrial SHP2 in regulating mtROS is context-dependent, as described above. While overactive SHP2 in mitochondria has been associated with increased mtROS production—leading to DNA damage, disruption of metabolic homeostasis, and activation of ERK1/2 signaling—SHP2 can also act to suppress excessive ROS generation under certain conditions, such as through the dephosphorylation of ANT1 to preserve mitochondrial membrane potential. These divergent effects highlight SHP2’s role in maintaining redox balance rather than simply promoting oxidative stress. When mitoROS levels are elevated, they can trigger the senescence-associated secretory phenotype (SASP), characterized by the expression and secretion of pro-inflammatory cytokines and matrix-degrading enzymes [36,37]. Thus, mitochondrial SHP2 may either inhibit or promote senescence depending on the cellular environment and upstream signals (Figure 5).

This dual role of SHP2 contrasts with its well-established anti-inflammatory functions in immune cells. In T cells and macrophages, SHP2 acts downstream of PD-1 signaling to promote immune tolerance by dampening T cell receptor signaling and limiting cytokine release. Moreover, SHP2 is essential for anti-inflammatory M2 macrophage polarization (Figure 4). These contrasting roles highlight the compartment- and context-specific nature of SHP2 function: while cytosolic SHP2 in immune cells may suppress inflammation, mitochondrial SHP2 in vascular or stromal cells may exacerbate inflammation and senescence under stress conditions (Figure 5).

In the cardiovascular system, this functional dichotomy is particularly relevant. Mitochondrial SHP2-driven senescence has been linked to endothelial dysfunction, impaired angiogenesis, cardiomyocyte hypertrophy, and accelerated atherosclerosis [38,39]. However, SHP2 can also act protectively by inhibiting NLRP3 inflammasome activation and limiting mtROS, thereby preserving mitochondrial integrity (Figure 5). These protective effects may be compromised during ICI therapy, which can inhibit SHP2 activity either directly or indirectly by disrupting upstream signaling. As a result, the loss of SHP2’s regulatory function may contribute to uncontrolled immune activation, oxidative stress, and cardiovascular toxicity [40]. These findings emphasize the importance of considering cell type specificity, subcellular localization, and therapeutic context when interpreting SHP2 function—and when targeting SHP2 in both cancer and vascular disease [41,42,43] (Figure 5).

## 3. SHP2 in Cancer and Cancer Treatments

The oncogenic potential of SHP2 further validated its role as a growth-promoting effector. Gain-of-function (GOF) mutations in *PTPN11*, which encodes SHP2, lead to the constitutive activation of downstream RAS–ERK pathways that drive a myriad of developmental disorders and malignancies. Canonical activating mutations (e.g., D61G, E76K) have been associated with juvenile myelomonocytic leukemia (JMML), Noonan syndrome (NS), and myeloproliferative neoplasms (MPNs) [44,45,46]. In solid tumors, SHP2 is increasingly recognized as a common node of vulnerability. GOF mutations and overexpression have been shown to play a role in colorectal, breast, lung, and thyroid cancers, as well as in melanoma and neuroblastoma [44,47].

From the perspective of therapeutic intervention, pharmacologic inhibition of SHP2 has provided functional validation of its required role in oncogenic RTK signaling. SHP099, an allosteric SHP2 inhibitor, inhibited ERK activation in EGFR- and FGFR-dependent tumor cells in vitro and inhibited tumor growth in vivo, providing strong evidence that SHP2 is an obligate mediator of RTK–RAS signaling [48]. Collectively, these data support a model in which SHP2 is an obligate effector of growth factor signaling in physiological and pathological conditions (Figure 3).

## 4. The Potential Roles of SHP2 in T Cells, Myeloid Cells, and Endothelial Cells, Leading to ICI-Induced CVD: Insights from Animal Models

### 4.1. T Cells

PD-1 was first identified as a negative regulator of T cell activation in 1992 by Ishida et al. [42], and its role as a negative regulator was found to be linked to programmed death-ligand 1 (PD-L1) leading to the development of immune checkpoint blockade therapies [49]. The PD-1/PD-L1 axis was soon recognized as a valuable mechanism for tumor evasion of immune surveillance where T cell suppression leads to immune evasion [11]. Upon binding with PD-L1, PD-1 is phosphorylated at specific tyrosine residues, in the cytoplasmic immunoreceptor tyrosine-based switch motif (ITSM) and immunoreceptor tyrosine-based inhibitory motif (ITIM) [11,50]. Phosphorylation allows the protein SHP2 to be recruited to the receptor. The phosphorylated ITSM binds to the C-SH2 domain, while the phosphorylated ITIM binds to the N-SH2 domain to activate SHP2 (Figure 1 and Figure 2). Once recruited and activated, SHP2 will dephosphorylate proximal components of T cell receptor (TCR) signaling (like CD3ζ, ZAP70, PI3K), suppressing downstream cascades (like PI3K-AKT or Ras-ERK), depleting cytokine production (in particular, IL-2), and enforcing T cell anergy [51,52,53]. The SHP2-driven, immune suppressing cascade allows tumor cells to escape immune response and evade immune surveillance [53] (Figure 6).

SHP2, classically described as a positive mediator of proliferation and survival through Ras-MEK-ERK and PI3K-AKT pathways in non-immune lineages like endothelial and myeloid cells [10,54,55], is paradoxically a suppressive regulator in T cells [56]. In the instance of recruited SHP2 by inhibitory receptors (like PD-1 and CTLA-4), the suppressive role of SHP2 occurs by damping down TCR signaling and subsequent IL-2 production, while inhibiting the activation of downstream pathways (including NF-κB, NFAT, and mTOR) [17]. This is accomplished, in part, by SHP2 lowering the phosphorylation of CD28 and decreasing co-stimulatory engagement (via PI3K). Other inhibitory receptors (B- and T- lymphocyte attenuator, BTLA and T cell immunoglobulin and ITIM domain, TIGIT) also recruit SHP2 through ITIM domains that contribute to the maintenance of T cell exhaustion [17,18].

One reason is that PD-1 engagement pulls SHP2 away from its active substrates in the Ras/ERK pathway to diminish its ability to promote T cell activation and favors signal inhibition [57]. The evidence indicates that SHP2 function in T cells is contextual and in any given moment, SHP2 can promote or inhibit activation depending on the receptors it has engaged and the signaling context through which it has engaged T cells. This behavior is flexible and stimulus-dependent, thus allowing SHP2 to act as a master regulator of T cell function; consequently, this provides a unique opportunity to study SHP2 with conditional genetic-engineering approaches in animals. Conditional deletion of SHP2 in T cells with CD4-Cre (Ptpn11fl/fl; CD4-Cre) is providing evidence that SHP2 is not required for T cell exhaustion: Ptpn11-deficient mice in chronic viral infection or cancer/melanoma challenge did not display differing expressions of T cell exhaustion marker expressions and responses to anti-PD-1 treatment as compared to the wild-type or externally wild-type mice, suggesting that alternative phosphatases (e.g., SHP1) can mediate inhibitive signals in vivo [58]. The findings in the absence of Ptpn11 in mice expressing constitutively active SHP2 D61Y mutant, in conjunction with Lck-Cre or pTα-Cre, showed increased expression of PD-1 on CD4+ and CD8+ T cells, diminished proliferation, increased apoptosis, and altered memory T cell populations with increased effector memory and intact immune clearance of viral pathogens [59].

SHP1 (*PTPN6*) and SHP2 (*PTPN11*) are nearly identical family members in the class of tyrosine phosphatases, with the capability to regulate immune signaling based on binding phosphorylated ITIM/ITSM, with partially redundant and distinctly different functions. Using T cell-specific knockout models, it was found that both SHP1 and SHP2 redundantly constrained the differentiation of naïve T cells, predominately by SHP1; SHP2 loss alone enhanced anti-PD-1 therapy efficacy; however, dual loss of SHP1 and SHP2 leads to impaired tumor control as a result of CD4+ T cell activation-dependent cell death [60]. The dual contribution of SHP2 dysregulating tumor cell signals and inhibiting T cell function makes it a promising immuno-therapeutic target; for example, treatment of SHP2 inhibitor SHP099 was unchanged in tumor intrinsic model systems; however, it provided robust tumor burden reduction in immune composition mice through the induction of CD8+ T cell responses and increased expression of cytotoxic gene signatures. Moreover, SHP099 immuno-therapeutics in conjunction with anti-PD-1 therapy was more effective than either modality taken alone in models of colon cancer, suggesting that they have complementary effects [61]. Collectively these data support SHP2 as an important regulatory node in tumor immunity and a promising immuno-therapeutic agent that can be used to augment immune checkpoint inhibitor therapies.

### 4.2. Myeloid Cells

Beyond its role in the inhibition of T cells, PD-1/SHP2 signaling has a significant impact on the development and function of myeloid cells. GM-CSF signaling promotes PD-1 activation via phosphorylation and promotes the recruitment of the PD-1–SHP2 complex by the receptor. Inhibition of PD-1 or SHP2 translates to increased downstream transcription factors, homeobox A10 (HOXA10) and interferon regulatory factor 8 (IRF8), that are essential for myeloid differentiation and commitment to the monocytic dendritic lineage. It has been reported that PD-1/SHP2 can act as a negative regulator of myelocyte differentiation and forms a myeloid environment to inhibit anti-tumor immunity [5] (Figure 6B).

In the tumor microenvironment (TME), macrophages are regulated by tumor-derived CSF-1, which binds to CSF-1R and promotes their survival and polarization into tumor-promoting phenotypes [62]. These tumor-associated macrophages (TAMs) facilitate tumor growth by suppressing immune responses and enhancing angiogenesis. SHP2, a tyrosine phosphatase activated downstream of CSF-1R, amplifies this signaling by promoting RAS/MAPK and PI3K/AKT pathways, thereby enhancing macrophage proliferation and polarization [61]. SHP2 may also contribute to increased CSF-1 or CSF-1R expression via ERK-dependent transcriptional regulation, forming a positive feedback loop that sustains TAM activation [10].

Additionally, immune checkpoint pathways such as CD47/SIRPα inhibit phagocytosis by delivering a “don’t eat me” signal from tumor cells to macrophages. Li et al. [55] reported that SIRPα on tumor-infiltrating macrophages (TIMs) binds to CD47 on tumor cells, initiating a downstream signaling cascade that requires the deneddylation of SHP2. SHP2 is constitutively neddylated at lysine residues K358 and K364, maintaining it in an autoinhibited state. Upon CD47 engagement, SIRPα recruits SHP2, which is deneddylated by SENP8, leading to its activation and subsequent dephosphorylation of substrates at the phagocytic cup, thereby inhibiting phagocytosis. Functionally, SHP2 neddylation inactivates its suppressive role and enhances the efficacy of colorectal cancer (CRC) immunotherapy. Furthermore, SHP2 allosteric inhibition sensitizes immunotherapy-resistant CRC, particularly in tumors characterized by high SIRPα and SHP2 expression with low NEDD8 levels. These findings identify SHP2 as a critical immune checkpoint regulator and a promising therapeutic target in CRC, and this SHP2-mediated suppression of efferocytosis allows tumor cells to evade immune clearance and persist within the TME, contributing to tumor progression and resistance to immunotherapy [63] (Figure 6B).

Blockade of SHP2 could shift TAMs toward the anti-tumor M1 phenotype to promote immune response and show SHP2 to be a strong fulfillment target for immunotherapy. Further, instead of enhancing M1, SHP2 contributes to an immunosuppressive TME and can diminish anti-tumor immunity by promoting M2 macrophage activation and dampening T cells’ response to ICIs. This suggests that myeloid cells have a role in therapy resistance and demonstrates that SHP2 can be a therapeutic target for ICI treatment [11,64,65]. ICI therapy can lead to potent, systemic immune activation with increased pro-inflammatory cytokines (IFN γ, TNF α, IL 6, and IL 1β) in a phenomenon sometimes described as a “cytokine storm”. While these mediators promote anti-tumor immunity, their elevated levels may promote off-target tissue injury and increased inflammation that can lead to (myocarditis, vasculitis) and accelerated atherosclerosis [66]. Preclinical models have provided valuable mechanisms linking ICI therapy to cardiovascular irAEs. For example, Pdcd1^−/−^ BALB/c [67] mice developed autoimmune dilated cardiomyopathy spontaneously via the presence of anti-cardiac troponin I autoantibodies, CD4+ and CD8+ T cell infiltration, and heart failure [68,69]. PD-1 deficiency on autoimmune-prone backgrounds such as *MRL-fas^lpr^* accelerates myocarditis and leads to fulminant cardiac inflammation [70,71].

The PD-1/PD-L1 immune checkpoint pathway was first implicated in vascular biology by Gotsman et al., who demonstrated that the genetic deletion of PD-1 or PD-L1 in Apoe^−/−^ mice resulted in significantly larger, more inflamed, and unstable atherosclerotic plaques [72]. This pivotal study identified PD-1 signaling as a critical negative regulator of T cell-driven vascular inflammation, acting as an “immune brake” to prevent excessive immune activation in the vessel wall. Subsequent research has shown that PD-1 plays a broader role in maintaining vascular homeostasis. In atherosclerosis-prone models, PD-1 blockade accelerates disease progression by enhancing T cell activation, monocyte recruitment, foam cell formation, and endothelial dysfunction—key processes in plaque development and destabilization [73,74,75]. In particular, both single cell RNA sequencing and mass cytometry approaches have mechanistically extrapolated that T cells, specifically cytotoxic CD8+ T cells and Th1-polarized CD4+ T cells, accumulate within atherosclerotic plaques (in both humans and mice) and these T cells contribute to necrotic core formation and plaque instability [76]. For example, in Ldlr^−/−^ mice, CD8+ T cells contribute to atherosclerosis by promoting early monopoiesis and macrophage cell death; upon CD8+ T cell depletion, atherosclerosis was lessened [72,73]. PD-1-deficient models likewise showed these effects, which included a predominant CD8+ T cell immune response signaling a form of cell necrotic pathway.

PD-1 signals through SHP2, a phosphatase that also regulates macrophage activation. Recent studies show that SHP2 inhibition or deletion in macrophages exacerbates atherosclerosis in ApoE^−/−^ and Ldlr^−/−^ mice by increasing inflammation, impairing efferocytosis, and destabilizing plaques. Mechanistically, SHP2 stabilizes PPARγ, a transcription factor essential for anti-inflammatory macrophage function, which is degraded upon SHP2 loss [77]. PD-1/SHP2 signaling also influences myeloid cell differentiation, as shown in tumor models where PD-1-SHP2 restrains GM-CSF-mediated activation of transcription factors like HOXA10 and IRF8 [5]. These findings suggest that PD-1/SHP2 signaling in macrophages plays a protective role in atherosclerosis by limiting inflammation and promoting resolution through PPARγ stabilization and regulated myeloid differentiation (Figure 6B).

Together these data suggest that PD-1/SHP2 signaling regulates myeloid differentiation, inflammatory tone, and vascular immune homeostasis. The disruption of PD-1/SHP2 signaling via genetic ablation or blockade of immune checkpoints adds to myelopoiesis, and macrophage polarization to pro-inflammatory macrophage, and pathological myeloid–T cell interaction leading to atherosclerosis, myocarditis, and vasculitis. Our understanding of these pathways in mouse models led to important insights and provided some framework by which we can predict, detect, and manage cardiovascular irAEs. Future studies of the dynamics of checkpoint signaling in tissue resident myeloid compartments will be important to enhance immunotherapies while preserving cardiovascular health.

### 4.3. Contextual Duality of SHP2 in Endothelial Inflammation: The Influence of Oxidative Stress

The deletion or silencing of SHP 2 specifically in ECs, or endothelial cells, increased the expression of adhesion molecules ICAM-1 and VCAM-1, increasing leukocyte adhesion and loss of endothelial barrier function, defining the features of early vascular inflammation [78]. In these settings, SHP 2 inhibits the association of ICAM-1 and VE cadherin-β catenin complexes, which provides a basis for regulating leukocyte transmigration and the dynamic re-assembly of the endothelial junctions [78]. Based on the soluble PD 1 receptors at the immune checkpoint directly recruiting SHP 2 through the specific motif ITIM/ITSM in the cytoplasmic tail, SHP 2 is an important signal transducer in PD 1 immune checkpoint signaling [11]. While PD-1 is more known for its immunosuppressive role on T cells, its function within ECs can be different.

SHP2 plays a central role in maintaining endothelial cell homeostasis by stabilizing the endothelial barrier and limiting inflammatory signaling under physiological conditions [78,79]. Through its regulation of junctional integrity and downstream signaling pathways, SHP2 helps preserve vascular quiescence and prevents leukocyte infiltration [79]. Therefore, in the presence of oxidative stress, such as hydrogen peroxide (H_2_O_2_) exposure, SHP2 inactivates and promotes inflammation due to the upregulation of the adhesion molecules ICAM-1 and VCAM-1, which promote monocyte adhesion to the endothelium [80]. Oxidative stress leads to H_2_O_2_-induced crosstalk between SHP2 and PP2A, which activates the crosstalk between SHP2 and PP2A and increases NF-κB signaling, which ultimately favors sustained endothelial–leukocyte crosstalk [81]. In inflammatory conditions, such as when ROS generation is induced from cholesterol crystals, the inactivation of SHP2 augments NF-κB-driven adhesion molecule gene transcription that drives vascular inflammation [82] (Figure 7A).

However, under pathological stress such as radiation exposure, SHP2 signaling may become maladaptive. Emerging evidence suggests that radiation can increase SHP2 activity, leading to endothelial dysfunction, increased permeability, and heightened inflammatory responses. In the clinically relevant radiation-induced lung injury (RILI) murine model characterized by progressive endothelial dysfunction and progressive fibrosis, SHP2 activity was measured to be elevated in ECs post thoracic irradiation. Endothelial-specific deletion of SHP2 resulted in reduced collagen deposition and dysregulated expression of Jagged 1 (Jag1), a Notch ligand shown to modulate macrophage polarization. Mechanistically, EC-derived Jag1 activates alternative macrophage polarization through Notch signaling in a paracrine fashion, creating a pro-fibrotic immune environment. This pathway was determined to be active in peripheral leukocytes from cancer patients following chest irradiation, supporting the potential relevance and clinical translational applicability. In summary, these results suggest that while SHP2 supports vascular stability and homeostasis in healthy tissues, the upregulation of SHP2 in vascular tissue in response to radiation stress may be detrimental by promoting vascular inflammation and fibrosis, supporting SHP2 as a context-dependent therapeutic target in radiation-induced vascular injury [83]. It is also important to consider that this pathogenic effect may occur via mitochondrial activation of SHP2 as a consequence of radiation damage, as detailed in Section 2.2, but a thorough characterization of this mechanism is still required (Figure 7A).

Wang et al. [41] demonstrated that PD-1/PD-L1 signaling plays a protective role in hypoxia-induced pulmonary hypertension (HPH), a condition driven by Th17-mediated inflammation. Their study showed that PD-1/PD-L1 expression is downregulated in HPH, particularly in endothelial cells, and that PD-1-deficient mice developed more severe disease under hypoxia. The restoration of PD-L1, either through recombinant protein or endothelial overexpression, alleviated HPH, while PD-L1 blockade enhanced endothelial angiogenesis, adhesion, and pyroptosis. Mechanistically, hypoxia-induced PD-L1 downregulation occurred via ubiquitination, and PD-L1 suppressed Th17 responses through the PI3K/AKT/mTOR pathway. Although the role of SHP2 in PD-1/PD-L1-mediated endothelial protection remains to be fully defined, these findings suggest that PD-1-dependent SHP2 signaling may contribute to endothelial stability. Importantly, this raises concern that ICIs, which block PD-1/PD-L1 interactions, could disrupt endothelial homeostasis and promote vascular dysfunction. Further investigation is needed to clarify the mechanisms by which PD-1/SHP2 signaling modulates endothelial responses in both physiological and pathological contexts.

### 4.4. Vascular Smooth Muscle Cells

SHP2 was recognized as a pro-atherogenic mediator in vascular smooth muscle cells (VSMCs) through its role in mitogenic signaling. For example, Chen et al. showed in Ldlr^−/−^ mice given a high-cholesterol diet that pharmacological blockade of SHP2 with phenylhydrazonopyrazolone sulfonate 1 (PHPS1) reduced ERK phosphorylation, inhibited VSMC proliferation, and reduced neointimal thickening. Together, these findings suggest that targeting SHP2 in VSMCs to modulate the SHP2–ERK signaling pathway may be an athero-protective strategy by potentially controlling pathological vascular remodeling [84] (Figure 7B).

### 4.5. Summary: Context-Dependent Roles of SHP2 in Atherosclerosis and ICI-Associated Vascular Diseases

SHP2 demonstrates varied and context-dependent effects on atherosclerosis that change quite substantially across different vascular and immune cell types (Table 1, Table 2 and Table 3). In macrophages, SHP2 is overall considered athero-protective, facilitating M2-like polarization and anti-inflammatory properties, which promote plaque stability. The inhibition of SHP2 or loss of SHP2 in macrophages has been associated with increased inflammation and foam cell formation, accelerating disease progress. In T cells, SHP2 functions in a PD-1-dependent manner, acting downstream of PD-1 to inhibit T cell receptor signaling, and limiting excessive immune activation. While this PD-1 function is beneficial to maintaining immune balance iatherosclerosis, chronic activation of PD-1 signaling via SHP2 can promote an exhausted T cell phenotype, further limiting anti-inflammatory resolution in atherosclerosis plaques.

In VSMCs, however, SHP2 functions as a pro-atherogenic effector, promoting and amplifying ERK/MAPK signaling pathways that promote proliferation, migration, and neointimal formation, all of which are key contributors during plaque formation. Importantly, pharmacological inhibition of SHP2 in VSMCs using PHPS1 showed reduced intimal hyperplasia and better plaque composition in mice. For endothelial cells, SHP2 also possesses a dual, context-dependent function in vascular inflammatory regulation. In contexts of oxidative stress (H_2_O_2_ and cholesterol crystals), SHP2 becomes inactivated, which promotes the upregulation of adhesion molecules including ICAM-1 and VCAM-1 and facilitates monocyte recruitment to the endothelium. In contrast, although exposure of endothelial cells to total body radiation activates SHP2, radiation increased inflammation by upregulating adhesion molecule expression and activation of downstream inflammatory pathways including Notch-Jag1 signaling, ultimately contributing to leukocyte infiltration, macrophage polarization, and fibrotic remodeling. Furthermore, VSMC and mitochondrial SHP2 may trigger pro-atherogenic effects. Collectively, these studies highlight the complex, stimulus-dependent nature of SHP2 signaling for endothelial biology where both suppression and activation can drive vascular inflammation and dysfunction depending on context. SHP2 is remarkable as a complex signaling hub and the atherosclerotic implications of SHP2 actions in each affected cell type are nuanced and context-dependent. This is important to address when developing cell type-specific targeted therapeutic strategies.

ICIs such as anti-PD-1 or anti-PD-L1 antibodies are utilized to block PD-1/SHP2 signaling in T cells to enhance anti-tumor immunity. While ICIs have been effective in treating a wide range of cancers, they not only block PD-1/SHP2 signaling in T lymphocytes, but also in endothelial cells and other non-immune cells. Removal of this regulatory mechanism by SHP2 may also lead to adjusted endothelial activation, increased leukocyte infiltration into tissues, and vascular inflammation. These effects, unintended, exacerbate atherogenesis, or can trigger cardiovascular irAEs, such as myocarditis, vasculitis, or plaque destabilization, which are increasingly being reported in patients receiving ICI therapy. Based on the variety of roles SHP2 plays in vascular biology, precise targeting of SHP2 for therapeutic purposes is needed. For instance, while the inhibition of SHP2 in VSMCs could have beneficial effects on reducing restenosis/neointimal formation, the inhibition of SHP2 in endothelial cells or macrophages could potentially exaggerate vascular inflammation. Future directions could look into cell type-specific targeting of SHP2, conditional knockout models, or local drug delivery systems. Furthermore, for patients undergoing ICI therapy as part of cancer treatment, cardiovascular monitoring may yield valuable opportunities for detection along with preservation strategies to preserve endothelial SHP2 signaling to mitigate long-term vascular adverse events. As the nexus of inflammation, immunity, and vascular disease, the modulation of SHP2, particularly in a cautiously crafted manner, may offer a chance to enhance oncological efficacy, while mitigating cardiovascular adverse events.

## 5. Clinical Relevance and Therapeutic Implications

Recent clinical trials have begun to evaluate SHP2 inhibitors, such as RMC-4630 and TNO155, in patients with RAS-driven malignancies. These pan-SHP2 inhibitors have demonstrated tolerable safety profiles and early signs of efficacy, particularly when used in combination with ICIs like anti-PD-1/PD-L1 antibodies [11,87]. Data from preclinical examinations imply that SHP2 inhibition can augment T cell activation and attenuate the resistance mechanism to ICIs, forming a justification for utilizing combination therapies. At this time, clinical data regarding combination studies are very scarce and further, cardiovascular safety has not been systematically evaluated, particularly in those with vascular disease [11].

Beyond its mechanistic role, SHP2 has significant clinical relevance. High SHP2 expression or activating mutations in *PTPN11* are associated with poor prognosis and aggressive disease phenotypes across multiple malignancies. In non-small cell lung cancer (NSCLC), elevated SHP2 activity correlates with advanced stage and reduced overall survival, particularly in tumors harboring KRAS or EGFR mutations [87,88]. Similarly, in breast cancer, SHP2 overexpression promotes HER2-driven oncogenic signaling and is linked to increased metastatic potential and poor clinical outcomes [89]. In hematologic malignancies, germline or somatic *PTPN11* mutations drive leukemogenesis in juvenile myelomonocytic leukemia (JMML) and acute myeloid leukemia (AML), where SHP2 activity predicts disease severity and relapse risk [90].

Therapeutically, SHP2 plays a critical role in resistance mechanisms. Elevated SHP2 activity often predicts resistance to targeted therapies such as EGFR or ALK inhibitors in NSCLC, as SHP2 sustains downstream MAPK signaling despite receptor blockade [91,92]. Preclinical studies demonstrate that SHP2 inhibition restores sensitivity to these agents, supporting its use in combination regimens [92]. Furthermore, SHP2 expression levels have been associated with response to ICIs. Tumors with high SHP2 activity may exhibit an immunosuppressive microenvironment, reducing ICI efficacy, whereas SHP2 inhibition enhances T cell activation and improves immunotherapy outcomes [11,41,93]. These findings position SHP2 as both a therapeutic target and a predictive biomarker for treatment response. Collectively, these associations underscore SHP2’s dual role in oncology: as a driver of tumor progression and as a determinant of therapeutic resistance and immunotherapy responsiveness. Its prognostic and predictive value highlights the need for SHP2 expression profiling in precision oncology strategies.

As previously discussed in Section 4, SHP2 exerts a dual influence on both endothelial cells and macrophages—two cell types that play pivotal roles in the pathogenesis of atherosclerosis. SHP2 contributes to endothelial integrity and redox balance, while also modulating macrophage-driven inflammatory responses. Given this dual impact, particularly in the context of ICI therapy, it is essential to implement cardio-oncology monitoring strategies. These should include non-invasive imaging to detect vascular inflammation, assessments of endothelial function, and biomarker surveillance for oxidative stress and cytokine activation. Such approaches are especially critical for patients with underlying cardiovascular disease, who may be more susceptible to vascular injury induced by combined ICI and SHP2 inhibitor therapies.

Mechanistically, SHP2’s redox-sensitive catalytic domain presents a pharmacological vulnerability that could be exploited for precision therapy. Redox-modulating agents, such as Nrf2 activators or thiol-reactive compounds, may allow for the context-dependent modulation of SHP2 activity, potentially reducing off-target effects [94,95]. Moreover, emerging strategies such as cell type-specific SHP2 targeting—via Proteolysis-Targeting Chimeras (PROTACs), nanoparticle delivery, or gene editing—offer promise in minimizing systemic toxicity [96,97]. Precision medicine approaches integrating genomic profiling, cardiovascular risk stratification, and biomarker-guided patient selection will be essential to safely advance SHP2-targeted therapies in oncology, particularly for patients with underlying cardiovascular disease.

## 6. Conclusions and Future Directions

SHP2 is a critical intracellular signaling mediator of PD-1/PD-L1 and is crucial in immune homeostasis and vascular function. In cancer, SHP2 has received a significant amount of attention not only for its role in PD-1-mediated inhibition of T cells, but also as a therapeutic target in its own right. Many scientific and clinical trials are currently investigating various combination therapies with ICIs and SHP2 inhibitors in an effort to overcome resistance and increase anti-tumor efficacy. The rationale is that by targeting tumor-intrinsic signaling as well as immunosuppression within the tumor microenvironment, treatments can amplify immune responses to cancer.

Immune checkpoints are receptors primarily expressed on immune cells that regulate immune homeostasis, with a significant impact on T cell function. Upon ligand binding, the PD-1 and CTLA-4 receptors become phosphorylated, which recruits the tyrosine phosphatase SHP2. SHP2 inhibits tyrosine phosphorylation-mediated signaling, thereby sup-pressing TCR and cytokine receptor-mediated T cell activation, which can contribute to tumorigenesis. ICIs counteract these pathways by inhibiting SHP2, promoting T cell activation through increased tyrosine phosphorylation. This process enhances immune surveillance, slows tumor growth, and boosts inflammation. Consequently, SHP2 inhibition is critical for the anti-tumor effects of ICIs, and the combination of ICIs with SHP2 inhibitors is under extensive clinical investigation. However, SHP2 also exhibit additional molecular functions beyond phosphatase activity—acting as a scaffold for SH2 domains and facilitating its translocation to mitochondria. These alternative functions exert both beneficial and detrimental effects on inflammation, proliferation, and ROS production, depending on the cell type, localization of SHP2, and the nature of the stimuli. As an SH2 domain scaffold, SHP2 interacts with signaling proteins such as GRB2, GAB1, and SOS, enhancing signaling efficiency and activating downstream pathways. In the absence of phosphatase activity, however, this scaffold function may lead to substrate trapping, thereby inhibiting downstream signaling. The activation of SHP2’s phosphatase function not only suppresses tyrosine phosphorylation-mediated signaling but also facilitates the release of these bound proteins, promoting productive downstream signaling. In mitochondria, SHP2 can inhibit ANT1 phosphorylation to reduce mtROS and inflammation, yet under certain conditions, mitochondrial SHP2 can enhance complexes I and III, elevating mtROS production and amplifying inflammation. Understanding these context-dependent roles is vital for interpreting the variability in ICI efficacy and cardiovascular complications observed among cancer patients.

SHP2 plays a context-dependent role in non-cancerous tissues, particularly within immune and vascular cells. Under physiological, non-oxidative conditions, SHP2 supports endothelial homeostasis by maintaining junctional integrity and suppressing inflammatory signaling. However, under oxidative stress—such as exposure to hydrogen peroxide or the presence of cholesterol crystals—SHP2 activity is inhibited, leading to the upregulation of adhesion molecules like ICAM-1 and VCAM-1, increased leukocyte adhesion, and endothelial activation. In contrast, under radiation exposure, SHP2 activation in endothelial cells contributes to lung injury and vascular fibrosis via NOTCH–Jag1-mediated recruitment of M2 (anti-inflammatory) macrophages. Additionally, SHP2 promotes atherosclerosis by enhancing ERK/MAPK signaling, which drives vascular smooth muscle cell proliferation, migration, and neointimal formation. These findings highlight SHP2’s dual role: protective in maintaining vascular stability under normal conditions, but potentially pathogenic under stress such as radiation-induced injury or in vascular smooth muscle cells. Therefore, while the inhibition of SHP2 may be advantageous in tumors, global inhibition of vascular SHP2 may result in exacerbated vascular inflammation or loss of protective signaling in healthy, non-cancerous tissues. Future therapeutics as a strategy for patients with comorbidities should be mindful to consider specific cell types and context (cardiovascular) modulation of SHP2 to allow them to achieve oncologic benefits post-ICI therapy, while not exposing them to cardiovascular harms.

## Figures and Tables

**Figure 1 antioxidants-14-01388-f001:**
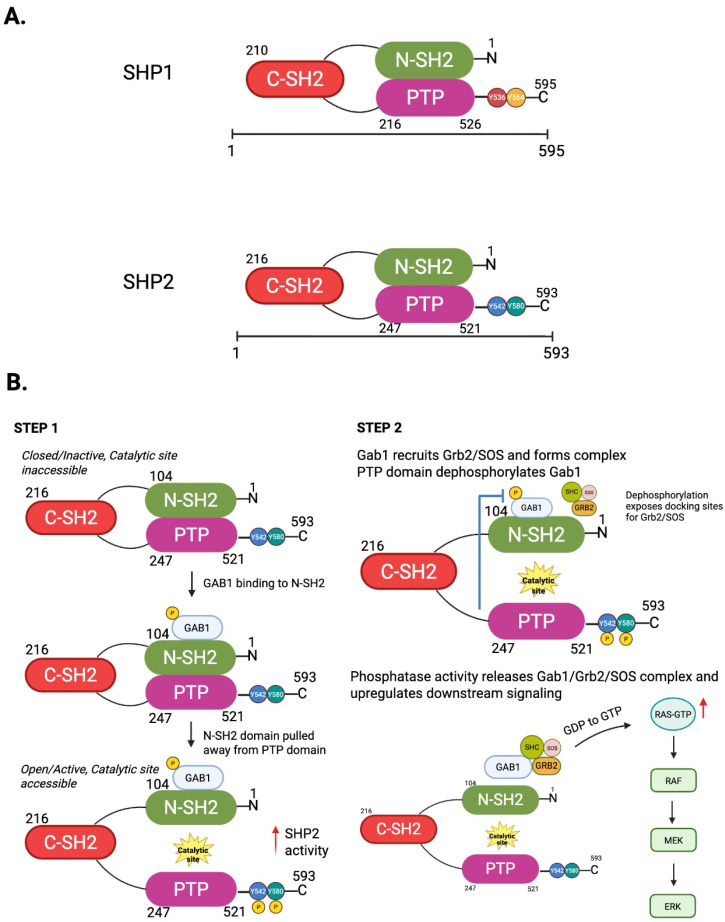
The two-step activation mechanism of SHP2 and its role as both a phosphatase and scaffold protein. (**A**) SHP1 and SHP2 are 68 kDa, multidomain enzymes that share similar sequences and functions. Both consist of two in-tandem Src Homology 2 domains (N-SH2 and C-SH2), a catalytic protein tyrosine phosphate domain (PTP), and a regulatory C-tail with main phosphorylation sites that vary for SHP1 (Y536 and Y564) and SHP2 (Y542 and Y580). More specifically, SHP1 is a 595 amino acid non-receptor protein tyrosine phosphatase that is encoded by the *PTPN6* gene. SHP1 is highly expressed in hematopoietic cells while its expression in other cells such as epithelial cells, astrocytes, and microglia can vary. Similarly, SHP2 is a 593 amino acid non-receptor protein tyrosine phosphatase that is encoded by the *PTPN11* gene. SHP2 is a broadly expressed cytoplasmic enzyme found in many different cell types. Both SHP1 and SHP2 exist in an autoinhibited basal conformation in which the N-SH2 domain tightly binds the PTP domain. Activating signals such as growth factors, cytokines, and tyrosine phosphorylated proteins like RTKs bind phosphopeptides to the C-SH2, with a conformational change occurring where N-SH2 releases from PTP domain and the enzyme is activated. (**B**) (Step 1) In its resting or quiescent state, the N-SH2 domain of SHP2 blocks its phosphatase domain (PTP), inhibiting access to catalytic site. When phosphotyrosine motifs such as adaptor protein GAB1 bind to the N-SH2 domain, the N-SH2 domain is pulled away from the C-terminal, which reveals the catalytic phosphatase domain. This conformational change allows SHP2 to adopt its catalytically active, open state. (Step 2) Once fully activated, SHP2’s PTP domain selectively dephosphorylates tyrosine residues on target substrates. This dephosphorylation event facilitates the release of signaling complexes such as GAB1/GRB2/SOS, thereby promoting downstream signaling cascades, including the activation of the RAS/MAPK pathway. Created in BioRender. Kim, J. (2026). https://BioRender.com/j1cdwrd.

**Figure 2 antioxidants-14-01388-f002:**
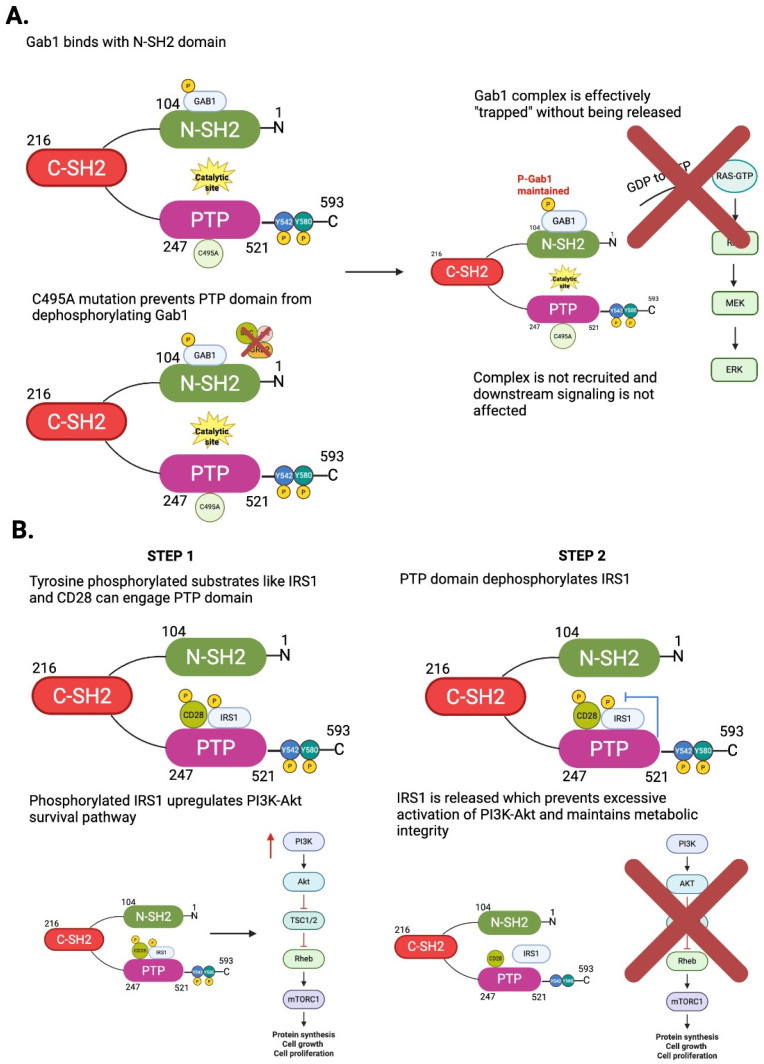
Substrate trapping effects caused by a phosphatase-deficient mutant and tyrosine-phosphorylated substrates that are directly associated with the tyrosine phosphatase domain, inhibiting downstream signaling. (**A**) Tyrosine-phosphorylated Gab1 associates with the N-terminal SH2 domain but cannot be dephosphorylated by the PTP domain due to its inactivity. As a result, tyrosine-phosphorylated Gab1 becomes trapped at the SH2 domain, preventing the activation of downstream GAB1/SOS signaling. (**B**) Certain tyrosine-phosphorylated substrates, such as CD28 and IRS1, directly bind to the PTP domain and are rapidly dephosphorylated by the SHP2 PTP domain. Consequently, excessive tyrosine phosphorylation of CD28 and IRS1 is reduced, and downstream signaling, including PI3-K and Akt activation, is inhibited. Created in BioRender. Kim, J. (2026) https://BioRender.com/xa47s2j.

**Figure 3 antioxidants-14-01388-f003:**
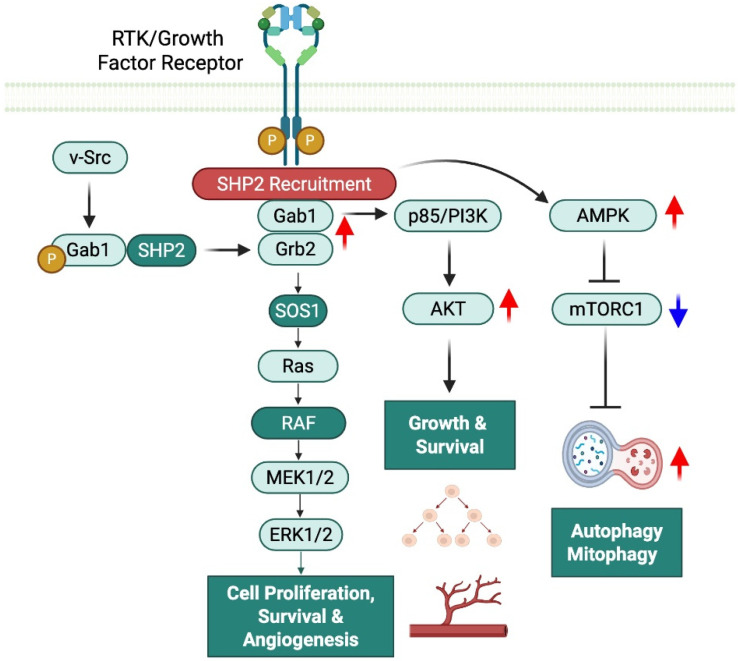
Positive effects of SHP2 in growth factor receptor tyrosine kinase signaling. The activation of growth factor receptor tyrosine kinases (RTKs) leads to the phosphorylation of intracellular components of the receptor, which in turn recruits SHP2 through its SH2 domain. The binding of SHP2 to RTKs facilitates the formation of complexes, such as GAB1/GRB2 and GAB1/p85/PI3-K, which subsequently activate downstream signaling pathways, including RAS/RAF/MAPK and AKT signaling. These pathways regulate critical cellular processes, including cell growth, survival, and angiogenesis. Additionally, the oncogene V-Src phosphorylates GAB1 and recruits SHP2, further promoting the GAB1/GRB2 signaling axis. This enhances MAPK signaling, leading to increased cell proliferation. SHP2 also upregulates AMP-activated protein kinase (AMPK), which plays a key role in regulating cellular metabolism. By activating AMPK, SHP2 promotes autophagy and mitophagy through the inhibition of mTORC1 signaling, contributing to cellular homeostasis and adaptation to stress. These findings highlight SHP2’s critical role in enhancing growth factor signaling and its impact on various cellular processes essential for tumorigenesis and tissue regeneration. Created in BioRender. Paniagua Bojorges, A. (2026) https://BioRender.com/u9fqpst.

**Figure 4 antioxidants-14-01388-f004:**
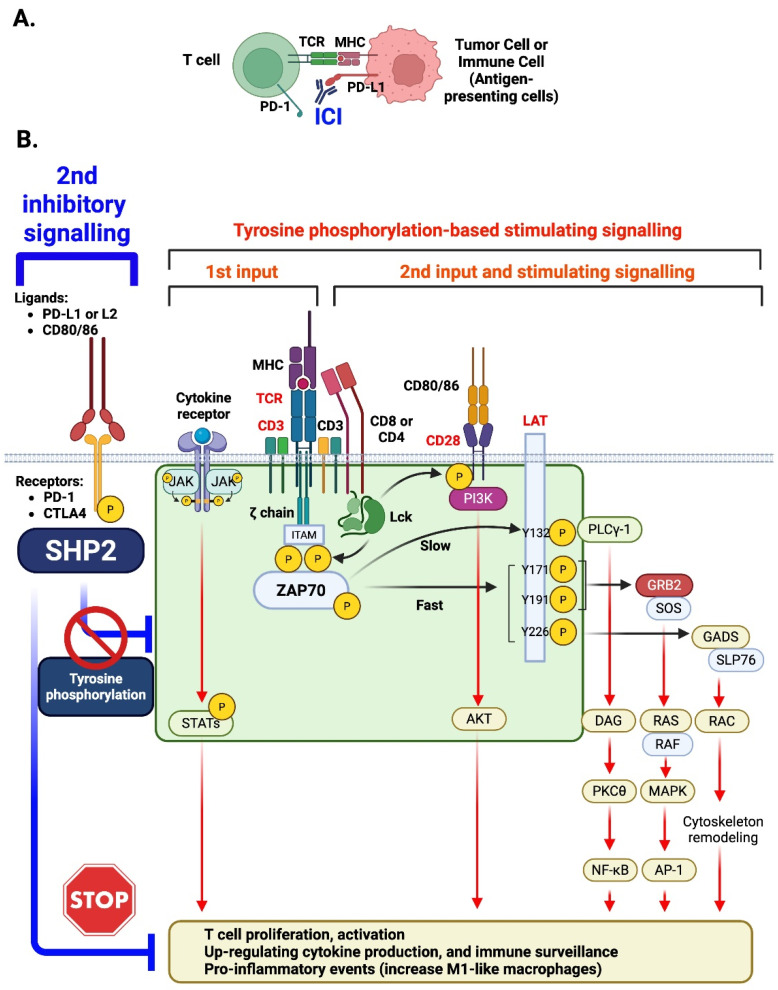
The role of SHP2 in T cell signaling and immune checkpoint inhibition. (**A**) The figure illustrates how immune checkpoint inhibitors (ICIs) enhance T cell-mediated anti-tumor responses. Normally, tumor cells express immune checkpoint ligands (such as PD-L1), which bind to immune checkpoint receptors (such as PD-1) on T cells. This interaction suppresses T cell activation, leading to immune evasion by the tumor. ICIs (represented as therapeutic antibodies in blue) block the binding between the checkpoint receptor and its ligand, preventing inhibitory signaling. As a result, T cell receptor (TCR) signaling and co-stimulatory signals are restored, leading to the reactivation of cytotoxic T cells. This reactivation promotes T cell proliferation, cytokine production, and direct tumor cell killing, thereby enhancing anti-tumor immunity. (**B**) This schematic outlines the signaling pathways involved in T cell activation and immune checkpoint inhibition. Upon engagement of cytokine receptors (CRs) and the T cell receptor (TCR), tyrosine phosphorylation-based signaling cascades are triggered, with key contributions from the TCR, CRs, and co-stimulatory signals (e.g., CD3, CD4). These signals activate tyrosine kinases such as Janus kinase (JAK) or Zeta-chain-associated protein kinase 70 (ZAP70), initiating downstream tyrosine phosphorylation events that regulate T cell activation and function. The figure further illustrates the activation of several downstream signaling pathways, including JAK/Signal Transducer and Activator of Transcription (STAT), Phosphoinositide 3-kinase/Protein Kinase B (PI3K/AKT), Phospholipase C gamma 1 (PLC-γ1), GRB2/SOS, and Ras-related C3 botulinum toxin substrate (RAC). These pathways play vital roles in promoting T cell proliferation, cytokine production, immune surveillance, and pro-inflammatory responses, including the activation of M1-like macrophages. SHP2 is crucial in regulating these signaling pathways. It negatively modulates the activation of T cells by dephosphorylating key components such as JAK/STATs, ZAP70, CD28, and Linker for activation of T cells (LAT). The inhibitory role of SHP2 is a key target for immune checkpoint inhibitors (ICIs), which block this secondary inhibition, thereby enhancing T cell activation and promoting a stronger immune response. Created in BioRender. Paniagua Bojorges, A. (2025) https://BioRender.com/8d0508f.

**Figure 5 antioxidants-14-01388-f005:**
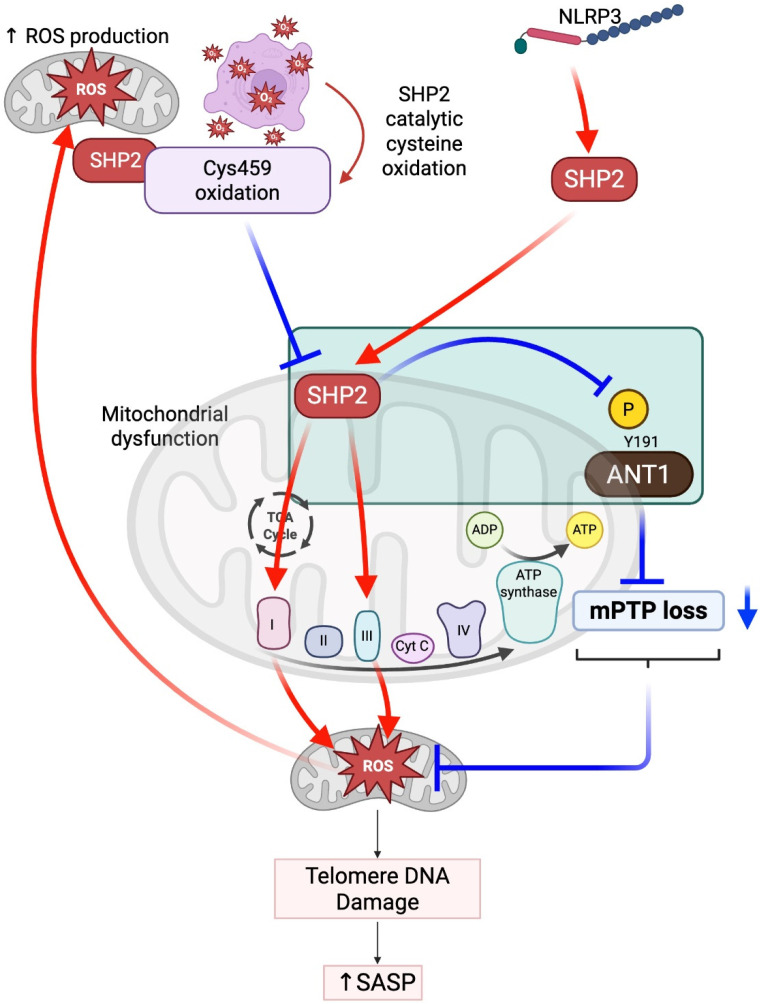
The role of mitochondrial SHP2 in mitochondrial ROS production and senescence. The activation of NLRP3 translocates SHP2 to the mitochondria, where it dephosphorylates ANT1 (adenine nucleotide translocator 1) at tyrosine 191 (Y191). This dephosphorylation activates ANT1 and promotes the loss of the mitochondrial permeability transition pore (mPTP), preserving mitochondrial function and integrity. However, ROS generated within the cell oxidize SHP2 at cysteine 459 (Cys459), which inhibits SHP2’s tyrosine phosphatase activity. As a result, the inhibition of mitochondrial SHP2 can trigger the loss of mPTP and promote mtROS production by upregulating ANT1 phosphorylation, creating a feedback loop that exacerbates mitochondrial dysfunction. Interestingly, it has also been reported that mitochondrial SHP2 can increase the activity of complexes I and III, further promoting mtROS production. This effect appears to be contradictory to SHP2’s role in regulating ANT1. Therefore, the role of SHP2 in mitochondria is context-dependent and warrants further investigation. The resulting increase in ROS levels contributes to telomere DNA damage, a critical factor in cellular aging. Telomere damage activates the senescence-associated secretory phenotype (SASP), which involves the secretion of pro-inflammatory cytokines. These processes accelerate cellular senescence, impairing cell function and promoting age-related diseases. This diagram highlights the complex, interconnected role of SHP2 in regulating mitochondrial function, oxidative stress, and telomere damage, all of which contribute to cellular aging and the development of age-associated pathologies. Created in BioRender. Paniagua Bojorges, A. (2026) https://BioRender.com/u9fqpst.

**Figure 6 antioxidants-14-01388-f006:**
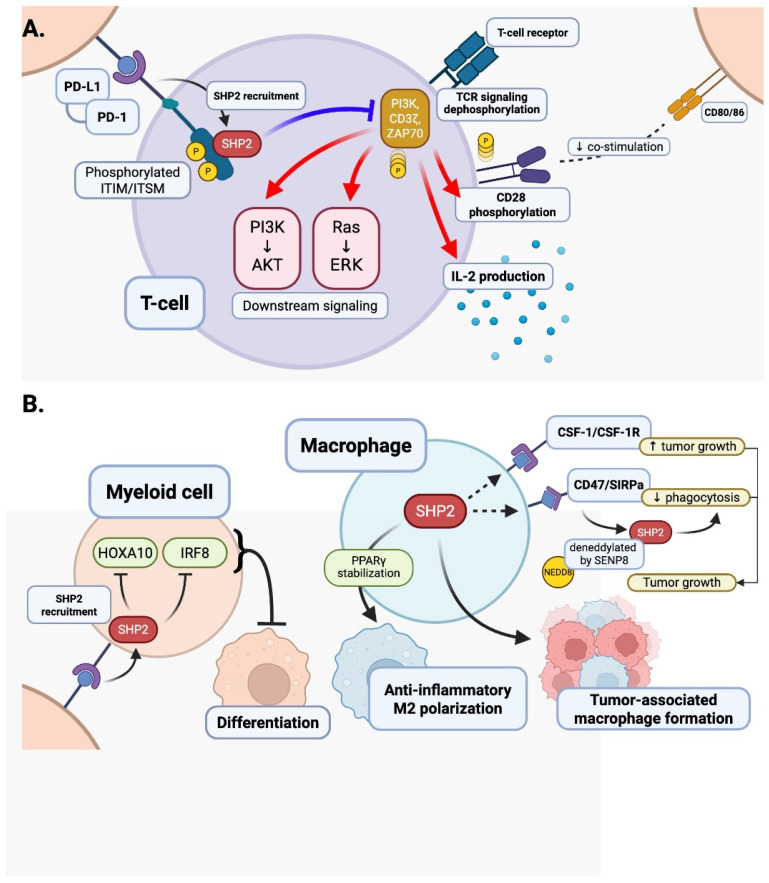
Cell-specific effects of SHP2 T cells and myeloid cells. (**A**) In T cells, PD-1 engagement by PD-L1 recruits SHP2 to phosphorylated ITIM and ITSM, leading to the dephosphorylation of TCR signaling components (CD3ζ, ZAP70, PI3K). This suppresses downstream pathways (PI3K-AKT, Ras-ERK), reduces IL-2 production, and decreases phosphorylation of co-stimulatory protein CD28, which in turn decreases co-stimulatory engagement. (**B**) In myeloid cells, PD-1–SHP2 signaling inhibits differentiation by reducing HOXA10 and IRF8, while in macrophages, it regulates CSF-1/CSF-1R and CD47/SIRPα pathways [5]. SHP2 promotes immunosuppressive M2 polarization through the stabilization of PPARg [54] and tumor-associated macrophage formation, thereby limiting anti-tumor immunity and contributing to resistance to checkpoint inhibitor therapy. SHP2 may also contribute to an increase in CSF-1 and CSF-1R expression. Additionally, upon CD47 engagement, SIRPα recruits SHP2, which is deneddylated by SENP8, leading to its activation and subsequent dephosphorylation of substrates at the phagocytic cup, thereby inhibiting phagocytosis. Created in BioRender. Pereda, C. (2026) https://BioRender.com/rbj0i0b.

**Figure 7 antioxidants-14-01388-f007:**
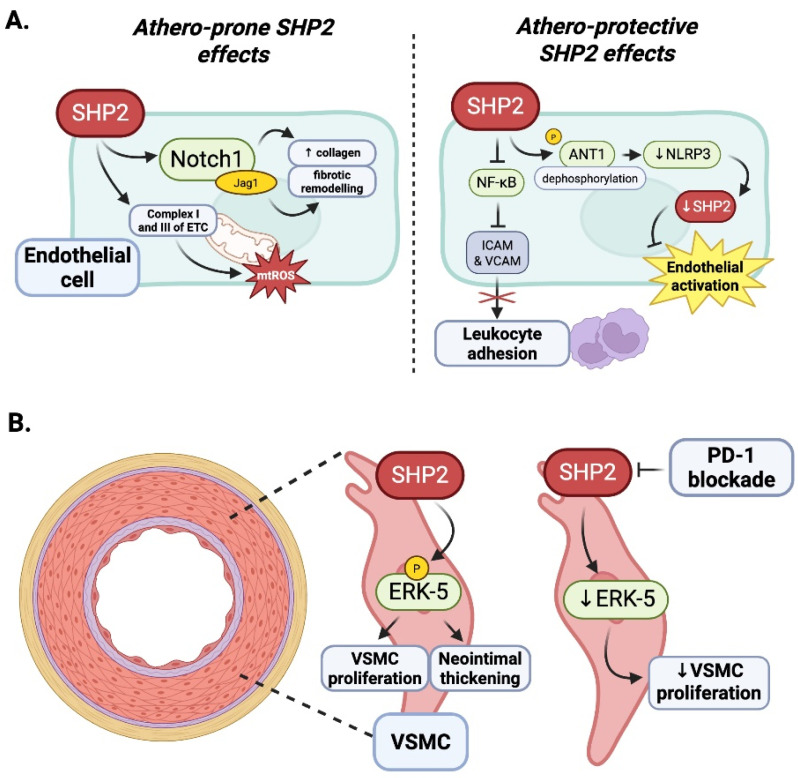
Cell-specific effects of SHP2: Endothelial cells and vascular smooth muscle cells. (**A**) In endothelial cells, SHP2 signaling can have positive and negative effects on atherosclerosis. SHP2 upregulation can be detrimental to tissues through fibrotic remodeling and collagen deposition by SHP2-induced dysregulation of Jag1, a Notch1 ligand [83]. Oxidative stress is also promoted by SHP2 by increased activity of complexes I and III in the ETC, which generates increased mtROS [33]. SHP2, however, can also dampen inflammatory response through reduced NF-κB signaling, which results in decreased leukocyte adhesion, as well as decreased phosphorylation of ANT1, resulting in the inhibition of NLRP3 inflammosome and decreased mtROS production, thus ameliorating endothelial activation and promoting athero-protective effects. (**B**) In vascular smooth muscle cells (VSMCs), SHP2 promotes mitogenic signaling via ERK phosphorylation, contributing to VSMC proliferation and neointimal thickening. Pharmacological inhibition of SHP2 reduces ERK activity, limits VSMC proliferation, and may provide athero-protective effects. Created in BioRender. Pereda, C. (2026) https://BioRender.com/xf2y3oi.

**Table 1 antioxidants-14-01388-t001:** SHP2 negative effects on signaling events.

Reference	SHP2 Effect	Signaling Pathways Regulated	Cellular Localization	ROS Involvement	Phosphatase Activity	Scaffold/Structural Role
Christofides et al., 2023, Nat Immunol [5]	Negative regulator of myeloid differentiation via PD-L1–SHP2; deletion boosts anti-tumor myelopoiesis	PD-1/ITSM → SHP2; GM-CSF/GM-CSFR; HOXA10/IRF8 axis	Plasma membrane (PD-1 complex), cytosol	Not assessed	Implicated/required (genetic loss phenocopies SHP2 absence)	PD-1 ITSM docking; GM-CSFR/LYN complex assembly
Wei et al., 2022, JMCB [32]	Negative regulator of AKT in ovary	PI3K/AKT (FSH/H_2_O_2_ context)	Cytosol; membrane-proximal	Upstream (H2O2 modulates SHP2 → p85)	Required (for AKT restraint)	PI3K p85 interaction
Guo et al., 2017, Nat Commun [35]	Negative regulator of inflammasome	NLRP3 via ANT1 mitochondrial homeostasis	Mitochondria	Downstream (limits mROS)	Required	ANT1-linked complex
Wang et al., 2021, Sci Rep [41]	SHP2 blockade ↑ anti-tumor immunity	RTK/MAPK, PD-1-related programs	Plasma membrane; cytosol	Not assessed	Inhibited (drug)	RTK/PD-1 complexes perturbed
Lan et al., 2015, EMBO J [42]	Suppresses senescence (pro-tumor)	Src–FAK–MEK/ERK → Skp2/AurA/DIII	Cytosol; membrane (functional)	Not assessed	Not directly tested	Src/FAK/MEK axis organization
Marasco et al., 2020, Sci Adv [52]	PD-1 directly activates SHP2	PD-1/ITIM/ITSM → SHP2; TCR inhibition	Plasma membrane (immune synapse)	Not assessed	Required	PD-1 tail ITIM/ITSM docking (structural)
Panchal et al., 2022, JACI [56]	Allosteric inhibition restores T cell function (SAP deficiency)	PD-1/SHP2-mediated negative signaling; TCR/SLAM	Immune synapse	Not assessed	Inhibited (drug)	PD-1/SAP pathway docking curtailed
Gavrieli et al., 2003, BBRC [18]	Negative (BTLA recruits SHP2)	BTLA/ITIM → SHP2; TCR inhibition	Plasma membrane (immune synapse)	Not assessed	Required	BTLA phosphotyrosine motifs
Yokosuka et al., 2012, J Exp Med [57]	Negative (PD-1 microclusters)	PD-1 microclusters–SHP2 → TCR inhibition	PD-1 clustered ITIM/ITSM platform	Not assessed	Required	PD-1 clustered ITIM/ITSM platform
Zhao et al., 2019, Acta Pharm Sin B [61]	SHP2 inhibition triggers immunity; synergy with PD-1 blockade	PD-1-related programs; RTK/MAPK	Cytosol; membrane	Not assessed	Inhibited (drug)	RTK immune complexes
Wu et al., 2024, ATVB [77]	Repressor of macrophage inflammatory activation	ROS/NLRP3 inflammasome restraint	Cytosol; mitochondria (macrophage)	Yes (limits ROS–NLRP3)	Required	Not specified
Yan et al., 2017, FASEB J [78]	Negative regulator of neutrophil adhesion; promotes transmigration	ICAM-1–VE-cadherin junction (conditional)	Endothelial junctions	Not assessed	Loss-of-function assays	Junctional complex dependent
Liu et al., 2022, iScience [83]	Endothelial SHP2 deletion exacerbates RILI via macrophage reprogramming	Notch signaling; alternate (via macrophage reprogramming)	Endothelium	Not assessed	Required	Not specified (specific case)

“→” = downstream/leads-to/promotes/results-in. “↑” = increase/upregulation / elevated level.

**Table 2 antioxidants-14-01388-t002:** SHP2 positive effects on signaling events.

Reference	SHP2 Effect	Signaling Pathways Regulated	Cellular Localization	ROS Involvement	Phosphatase Activity	Scaffold/Structural Role
Huang et al., 2002, JBC [12]	Positive (required for Elk-1 activation)	GAB1 → ERK → Elk1	Cytosol; plasma membrane	Not assessed	Required	GAB1 docking platform
Noguchi et al., 1994, Mol Cell Biol [24]	Positive (insulin Ras activation)	IR → IRS-1 → RAS/ERK	Plasma membrane; cytosol	Not assessed	Required	IRS-1/IR complex docking
Kandadi et al., 2010, Acta Pharmacol Sin [25]	Positive (migration/proliferation)	PDGF → ERK	Cytosol; plasma membrane	Not assessed	Required	PDGFR-Grb2 complex
Yu et al., 2014, Biochemistry [28]	GOF (LEOPARD) enhances activity	RAS/ERK hyperactivation	Cytosol; plasma membrane	Not assessed	Enhanced (mutants relieve autoinhibition)	Structural focus (no specific scaffold)
Wu et al., 2001, Oncogene [30]	Positive (growth-factor PI3K/AKT)	PI3K/AKT (and ERK interplay)	Membrane-proximal; cytosol	Not assessed	Required	GAB1, IRS-1 implied
Kan et al., 2024, JCI Insight [33]	GOF (E76K) activates complexes I and III	OXPHOS/ETC activation (metabolic)	Mitochondria (liquid–liquid phase separation/LLPS condensates)	Downstream (↑ mtROS)	Not required (LLPS-dependent)	LLPS-mediated ETC complex assembly
Bentires-Alj et al., 2004, Cancer Res [44]	GOF mutations (oncogenic)	Constitutive RAS/MAPK	Cytosol; plasma membrane	Not assessed	Required	Not scaffold-specific
Xu et al., 2013, PLoS One [45]	GOF → ↑ ROS; MPN	RAS/MAPK + ROS	Cytosol; mitochondria-linked	Yes (↑ ROS downstream of GOF)	Required	Not scaffold-specific
De Rocca-Serra-Nedelec et al., 2012, PNAS [46]	GOF hyperactivates ERK; IGF-1	GH → ERK hyperactivation; inhibition of IGF-1 secretion	Cytosol; plasma membrane	Not assessed	Required	GH receptor complex
Schneberger et al., 2014, Carcinogenesis [47]	GOF E76K drives lung tumors	ERK/MAPK hyperactivation	Cytosol; plasma membrane	Not assessed	Required	Not scaffold-specific
Fedida et al., 2018, Cancer Discov [48]	SHP2 enables adaptive resistance; inhibits feedback reactivation	RTK → RAS → ERK feedback regulation	Cytosol; plasma membrane	Not assessed	Required for reactivation prevention	GAB1/SOS-centered feedback node
Feng et al., 2021, ATVB [81]	Endothelial deletion ↓ VEGF/angiogenic signaling	VEGF–ERK engagement ↓ by SHP2 deletion	Cytosol; membrane-proximal	Yes (enhances NOX signaling)	Required	VEGFR2/Tie2 adaptor complex
Mattoon et al., 2004, BMC Biol [85]	Positive, anti-apoptotic	PI3K/AKT → caspase-3 suppression	Cytosol; membrane-proximal	Not assessed	Required	GAB1
Ushio-Fukai, 2006, Cardiovasc Res [86]	SHP2 → VEGF signaling → angiogenesis	VEGF/MAPK cascade	Endothelium (NOX microdomain)	Yes (enhances ROS)	Required	p47phox/NOX2 interactions

→ = leads to/results in/downstream step. ↑ = increase/upregulation/elevated signaling. ↓ = decrease/downregulation/reduced signaling.

**Table 3 antioxidants-14-01388-t003:** SHP2 dual effects on signaling events.

Reference	SHP2 Effect	Signaling Pathways Regulated	Cellular Localization	ROS Involvement	Phosphatase Activity	Scaffold/Structural Role
Xu et al., 2002, Exp Cell Res [7]	Context-dependent; mapping partners	Adhesion/immune: PZR, PECAM-1; growth: SHC, GAB1, IRS1	Cytosol; plasma membrane	Not assessed	Not directly tested	PZR, PECAM-1, SHC, GAB1, IRS1 as SHP2-binding scaffolds
Rota et al., 2018, Cell Rep [58]	Dispensable for PD-1 signaling/exhaustion in vivo	PD-1/TCR pathways (murine)	T cell membrane/cytosol (physiologic)	Not assessed	Not required (in vivo)	Not essential scaffold in that context
Foster et al., 2025, PNAS [60]	Dual SHP1/2 deletion → CD4+ ICOS, poor anti-tumor	TCR survival/apoptosis circuits	T cell compartments	Not assessed	Abolished (both PTPs)	Redundant checkpoint scaffolding removed
Canmann et al., 2022, Front Immunol [59]	GOF (constitutive) → ↑ memory T cell formation AND ↓ acute T cell activation	TCR/MAPK transcriptional programs	T cell compartments	Not assessed	Enhanced (constitutive)	Not primary (functional outcome focus)

→ = leads to/results in/downstream step. ↑ = increase/upregulation/elevated signaling. ↓ = decrease/downregulation/reduced signaling.

## Data Availability

No new data were created or analyzed in this study. Data sharing is not applicable to this article.

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
