# Peer review of "SHP2: A Redox-Sensitive Regulator Linking Immune Checkpoint Inhibitor Therapy to Cancer Treatment and Vascular Risk"

_antioxidants, 2025, doi:10.3390/antiox14121388_

Round 1
Reviewer 1 Report
This is a really excellent review presenting the complex role of SHP2 in metabolic regulation, demonstrating its involvement in immune checkpoint inhibitor therapy and vascular risk. It is a pleasure to read well-balanced explanations about the complex implications of this phosphatase in various signaling pathways, and I am convinced this manuscript deserves to be published. Congratulations to the authors!
I would recommend to refer to the uniprot entry name "Tyrosine-protein phosphatase non-receptor type 11", only after the Introduction the corresponding gene name PTPN11 is mentioned. SHP2 is the former designation, which is ok to be used but I think its helpful to immediately refer to its current designation. Otherwise I am really positively excited about this review, its accurate, it provides clear explanations and its really important and timely!
Author Response
Reviewer 1
“Major comments
This is a really excellent review presenting the complex role of SHP2 in metabolic regulation, demonstrating its involvement in immune checkpoint inhibitor therapy and vascular risk. It is a pleasure to read well-balanced explanations about the complex implications of this phosphatase in various signaling pathways, and I am convinced this manuscript deserves to be published. Congratulations to the authors!
Detailed comments
I would recommend to refer to the uniprot entry name "Tyrosine-protein phosphatase non-receptor type 11", only after the Introduction the corresponding gene name PTPN11 is mentioned. SHP2 is the former designation, which is ok to be used but I think its helpful to immediately refer to its current designation. Otherwise I am really positively excited about this review, its accurate, it provides clear explanations and its really important and timely!”
Thank you very much for your positive and encouraging feedback on our review. We truly appreciate your thoughtful comments and suggestions. In response to your recommendation, we have added the UniProt entry name “Tyrosine-protein phosphatase non-receptor type 11” along with the gene name PTPN11 in Lines 37–38 of the Abstract, as suggested. We agree that this improves clarity and consistency in nomenclature.
Thank you again for your valuable input and support.
Reviewer 2 Report
This review article summarizes current knowledge of SHP2 in cancer and cardiovascular diseases. Although the contents are informative and well-organized, the following issues must be addressed:
- Abstract: "VSMC" should be replaced with vascular smooth muscle cells
- Abstract: The aim of the review article should be clearly stated, like lines 127–130. It is unclear from the abstract which aspects of SHP2 the authors intend to review.
- Abstract: It should be mentioned that there is an association between ICIs/irAEs and cardiovascular complications.
- The legend is unnecessary for graphical abstract.
- line 108: Although some CD80/CD86 inhibitors are used to treat B-cell lymphoma, it is not certain that they are ICIs. Couldn't LAG-3 or TIM-3 inhibitors be more appropriate?
- lines 318–320: It is hard to understand why STAT1 is dephosphorylated in SHP2 deficient cells.
- Section 1.1: This section should be shortened because most references are somewhat outdated and the information is already summarized elsewhere.
- There are some typos and undefined abbreviations. For instance, line550–551: did not display differing 'expression' of T cell exhaustion marker 'expressions', line 742: PHPS1, line 756: iatherosclerosis... Text should be thoroughly double-checked.
- Although the physiological and pathological roles of SHP2 are well described, information regarding the clinical relevance of SHP2, such as its association with prognosis, disease severity, and therapeutic response, is relatively scarce. Please include any available information from previous studies in the manuscript.
Author Response
Reviewer 2
Major comments
“This review article summarizes current knowledge of SHP2 in cancer and cardiovascular diseases. Although the contents are informative and well-organized, the following issues must be addressed:”
Thank you for your thoughtful feedback and for recognizing the organization and informativeness of the review. We appreciate your comments and will carefully address each of the issues you have outlined. Our goal is to ensure that the revised manuscript meets the highest standards of clarity, completeness, and scientific rigor.
Detailed comments
- Abstract: "VSMC" should be replaced with vascular smooth muscle cells
Thank you for pointing this out. We have replaced “VSMC” with “vascular smooth muscle cells” in the Abstract as suggested.
- Abstract: The aim of the review article should be clearly stated, like lines 127–130. It is unclear from the abstract which aspects of SHP2 the authors intend to review.
Thank you for this valuable suggestion. We have revised the Abstract to clearly state the aim of the review, aligning it with the scope described in lines 127–130. The updated Abstract now specifies that the review focuses on SHP2’s biological functions, its role in cancer and cardiovascular disease, and emerging therapeutic strategies targeting SHP2.
We added the following in line 44-47:
This review aims to summarize current knowledge on SHP2/PTPN11 biology, its role in immune regulation, cancer progression, and vascular homeostasis, and to discuss emerging therapeutic strategies targeting this pathway.
- Abstract: It should be mentioned that there is an association between ICIs/irAEs and cardiovascular complications.
Thank you for this important suggestion. We have revised the Abstract to include a statement noting the association between immune checkpoint inhibitors (ICIs), immune-related adverse events (irAEs), and cardiovascular complications. This addition emphasizes the clinical relevance of SHP2 in the context of cardio-oncology.
We added the following in line 41-44:
Importantly, there is an association between immune checkpoint inhibitors (ICIs), immune-related adverse events (irAEs), and cardiovascular complications, underscoring the need to understand SHP2’s role in these processes.
- The legend is unnecessary for graphical abstract.
Thank you for your observation. We included the legend for the graphical abstract based on guidance from the journal office, which confirmed that it was acceptable. If the editorial team prefers its removal, we are happy to comply.
- line 108: Although some CD80/CD86 inhibitors are used to treat B-cell lymphoma, it is not certain that they are ICIs. Couldn't LAG-3 or TIM-3 inhibitors be more appropriate?
Thank you for your observation. We agree with your point and have corrected the text accordingly. The revised version now refers to LAG-3 and TIM-3 inhibitors as examples of ICIs in line 118-120, which are more appropriate in this context.
- lines 318–320: It is hard to understand why STAT1 is dephosphorylated in SHP2 deficient cells.
Thank you for pointing this out. We have corrected the text to clarify that SHP2 deficiency leads to prolonged STAT1 phosphorylation, not dephosphorylation. The revised sentence now accurately reflects the mechanism described in the cited studies in line 339-340.
- Section 1.1: This section should be shortened because most references are somewhat outdated and the information is already summarized elsewhere.
Thank you for your suggestion. We have shortened Section 1.1 as recommended by removing redundant details, while retaining essential background information for context.
- There are some typos and undefined abbreviations. For instance, line550–551: did not display differing 'expression' of T cell exhaustion marker 'expressions', line 742: PHPS1, line 756: iatherosclerosis... Text should be thoroughly double-checked.
Thank you for noting these issues. We have thoroughly reviewed the manuscript, corrected the typos, clarified undefined abbreviations, and fixed the examples you highlighted (lines 550–551, 742, and 756). The text has been double-checked to ensure accuracy and consistency throughout.
- Although the physiological and pathological roles of SHP2 are well described, information regarding the clinical relevance of SHP2, such as its association with prognosis, disease severity, and therapeutic response, is relatively scarce. Please include any available information from previous studies in the manuscript.
Thank you for this important suggestion. We have added detailed information regarding the clinical relevance of SHP2, including its association with prognosis, disease severity, and therapeutic response, based on previous studies. These additions are now included in the revised manuscript with appropriate references.
We added the following in line 918-940:
Beyond its mechanistic role, SHP2 has significant clinical relevance. High SHP2 expression or activating mutations in PTPN11 are associated with poor prognosis and aggressive disease phenotypes across multiple malignancies. In non-small cell lung cancer (NSCLC), elevated SHP2 activity correlates with advanced stage and reduced overall survival, particularly in tumors harboring KRAS or EGFR mutations [85,86]. Similarly, in breast cancer, SHP2 overexpression promotes HER2-driven oncogenic signaling and is linked to increased metastatic potential and poor clinical outcomes [87]. In hematologic malignancies, germline or somatic PTPN11 mutations drive leukemogenesis in juvenile myelomonocytic leukemia (JMML) and acute myeloid leukemia (AML), where SHP2 activity predicts disease severity and relapse risk [88].
Therapeutically, SHP2 plays a critical role in resistance mechanisms. Elevated SHP2 activity often predicts resistance to targeted therapies such as EGFR or ALK inhibitors in NSCLC, as SHP2 sustains downstream MAPK signaling despite receptor blockade[89] [90]. Preclinical studies demonstrate that SHP2 inhibition restores sensitivity to these agents, supporting its use in combination regimens [90]. Furthermore, SHP2 expression levels have been associated with response to ICIs. Tumors with high SHP2 activity may exhibit an immunosuppressive microenvironment, reducing ICI efficacy, whereas SHP2 inhibition enhances T-cell activation and improves immunotherapy outcomes[11,41,91]. These findings position SHP2 as both a therapeutic target and a predictive biomarker for treatment response. Collectively, these associations underscore SHP2’s dual role in oncology: as a driver of tumor progression and as a determinant of therapeutic resistance and immunotherapy responsiveness. Its prognostic and predictive value highlights the need for SHP2 expression profiling in precision oncology strategies.
Round 2
Reviewer 2 Report
Thank the authors for adequately addressing my comments and implementing most of the requested revisions.
The manuscript is now clearer and more accurate.